# Fat-to-Muscle Ratio Is Independently Associated with Hyperuricemia and a Reduced Estimated Glomerular Filtration Rate in Chinese Adults: The China National Health Survey

**DOI:** 10.3390/nu14194193

**Published:** 2022-10-08

**Authors:** Huijing He, Li Pan, Dingming Wang, Feng Liu, Jianwei Du, Lize Pa, Xianghua Wang, Ze Cui, Xiaolan Ren, Hailing Wang, Xia Peng, Jingbo Zhao, Guangliang Shan

**Affiliations:** 1Department of Epidemiology and Statistics, Institute of Basic Medical Sciences, Chinese Academy of Medical Sciences & School of Basic Medicine, Peking Union Medical College, Beijing 100005, China; 2Department of Chronic and Noncommunicable Disease Prevention and Control, Guizhou Provincial Center for Disease Control and Prevention, Guiyang 550004, China; 3Department of Chronic and Noncommunicable Disease Prevention and Control, Shaanxi Provincial Center for Disease Control and Prevention, Xi’an 710054, China; 4Department of Chronic and Noncommunicable Disease Prevention and Control, Hainan Provincial Center for Disease Control and Prevention, Haikou 570203, China; 5Department of Chronic and Noncommunicable Disease Prevention and Control, Xinjiang Uyghur Autonomous Region Center for Disease Control and Prevention, Urumqi 830001, China; 6Institute of Biomedical Engineering, Chinese Academy of Medical Sciences, Tianjin 300192, China; 7Department of Chronic and Noncommunicable Disease Prevention and Control, Hebei Provincial Center for Disease Control and Prevention, Shijiazhuang 050000, China; 8Department of Chronic and Noncommunicable Disease Prevention and Control, Gansu Provincial Center for Disease Control and Prevention, Lanzhou 730000, China; 9Department of Chronic and Noncommunicable Disease Prevention and Control, Inner Mongolia Autonomous Region Center for Disease Control and Prevention, Baotou 014000, China; 10Department of Chronic and Noncommunicable Disease Prevention and Control, Yunnan Provincial Center for Disease Control and Prevention, Kunming 650022, China; 11School of Public Health, Harbin Medical University, Harbin 150081, China

**Keywords:** fat-to-muscle ratio, body composition, uric acid, estimated glomerular filtration rate, mediation effect, risk factors

## Abstract

Background: The effects of the fat-to-muscle ratio (FMR) on hyperuricemia and a reduction in the estimated glomerular filtration rate (eGFR) are still unclear. Methods: Data from the China National Health Survey were used to explore the associations of the FMR with hyperuricemia and reduced eGFR. The fat mass and muscle mass were measured through bioelectrical impedance analysis. Mediation analysis was used to estimate the mediated effect of hyperuricemia on the association between the FMR and reduced eGFR. Results: A total of 31171 participants were included. For hyperuricemia, compared with the Q1 of the FMR, the ORs (95% CI) of Q2, Q3 and Q4 were 1.60 (1.32–1.95), 2.31 (1.91–2.80) and 2.71 (2.15–3.43) in men and 1.91 (1.56–2.34), 2.67 (2.12–3.36) and 4.47 (3.40–5.89) in women. For the reduced eGFR, the ORs (95% CI) of Q2, Q3 and Q4 of the FMR were 1.48 (1.18–1.87), 1.38 (1.05–1.82) and 1.45 (1.04–2.04) in men aged 40–59, but no positive association was found in younger men or in women. Hyperuricemia mediated the association between the FMR and reduced eGFR in men. The OR (95% CI) of the indirect effect was 1.08 (1.05–1.10), accounting for 35.11% of the total effect. Conclusions: The FMR was associated with hyperuricemia and reduced eGFR, and the associations varied based on sex and age. The effect of the FMR on the reduced eGFR was significantly mediated by hyperuricemia in men.

## 1. Introduction

Hyperuricemia, which is defined as elevated serum uric acid (SUA) when the SUA production exceeds its excretion, has been found to be associated with various health outcomes [1,2]. Data from the China National Health Survey (CNHS) revealed an almost 20% prevalence of hyperuricemia among adults in mainland China [3]. Hyperuricemia was also suggested to be one of the most important risk factors for kidney dysfunction [4,5,6], which has received increasing attention as a global public health issue.

People with kidney dysfunction are more likely to develop end-stage renal disease and cardiovascular disease and have a higher risk of mortality [5,7,8]. Elevated SUA was reported to be associated with kidney dysfunction, and previous epidemiological studies and basic science suggest that hyperuricemia is an important risk factor for the development of chronic kidney disease (CKD) [1].

As one of the shared risk factors for hyperuricemia and kidney dysfunction, excess adiposity was demonstrated to have a significant role in elevating the SUA in the general Chinese population in our previous studies [3,9,10]. Although plenty of adiposity measurements, such as the body mass index (BMI), body fat percentage, waist circumference, waist-to-height ratio, fat mass index, etc., have been applied in clinical practice as indicators of chronic disease, the common limitation of these indicators is that they only consider the effect of fat, while ignoring the skeletal muscle, which is another important tissue for maintaining metabolism health. The fat-to-muscle ratio (FMR), therefore, has been proposed as an alternative approach for assessing excess adiposity [11]. In recent years, researchers have begun to explore the value of the FMR for the identification of members of populations at a high risk of developing cardiometabolic disorders, such as type-2 diabetes and metabolic syndrome, and they found that the FMR is positively associated with an increased risk of such outcomes [11,12,13]. However, the associations of the FMR with hyperuricemia and reduced kidney function, both of which are important risk factors for cardiovascular disease and even death [1,8], are still unclear. Furthermore, as excess adiposity is considered as a common risk factor for hyperuricemia and kidney dysfunction, and hyperuricemia is strongly associated with kidney dysfunction, we hypothesized that there could be a potential mediating role of hyperuricemia in the pathway from FMR to reduced kidney function (which is termed as an indirect effect).

The glomerular filtration rate (GFR) is the ideal overall index of kidney function, but the measurement is cumbersome and impractical for the general detection and management of kidney disease [14]. In this case, the estimated GFR (eGFR) acquired by equations is widely used as a surrogate of GFR, especially in large-scale epidemiological studies. Here, using data from the CNHS, a nationwide cross-sectional study conducted in mainland China, we aimed to explore the role of FMR in, and its associations with, hyperuricemia and reduced eGFR in Chinese adults, and we further explored the possible mediating role of hyperuricemia in the association between the FMR and reduced eGFR.

## 2. Materials and Methods

### 2.1. Study Population

The CNHS was conducted from 2012 to 2017 by the Institute of Basic Medical Sciences, Chinese Academy of Medical Sciences. The study protocol has been published previously [15]. Briefly, using a multi-stage, stratified cluster sampling method, the CNHS recruited representative adults from eleven provinces in mainland China. In the sampling procedure, the geographic regions, degree of urbanization and economic development level, assessed according to the local GDP, were considered.

The inclusive criteria for the study population were an age of 20–80 years and the status of having lived in the local area for at least one year. The exclusive criteria meant that people with severe mental or physical disease, pregnant women or military personnel in active service were discounted. In this study, we used data restricted to Han Chinese individuals so as to reduce potential heterogeneity in terms of genetic or socio-cultural backgrounds.

The study was carried out in accordance with the Declaration of Helsinki. Ethical approval was obtained from the Bioethical Committee of the Institute of Basic Medical Sciences, Chinese Academy of Medical Sciences (No.029-2013). All participants provided written informed consent before the survey.

### 2.2. Data Collection and Measurements

A standardized questionnaire was designed to collect demographic and health-related information through face-to-face interview. All staff underwent a training program to guarantee their capability of conducting precise data collection and measurements. Information on demographic characteristics (sex, date of birth, permanent address, educational level), health-related lifestyle (alcohol drinking, cigarette smoking) and personal disease history (hypertension, diabetes, cardiovascular disease) were collected during the survey.

The standing height was measured to the nearest 0.1 cm using a fixed stadiometer. The weight, fat mass and muscle mass were measured through bioelectrical impedance analysis (BIA) (TANITA BC-420, Tanita, Tokyo, Japan). The average value of three blood pressure (BP) readings was recorded using a digital BP measurement device (Omron HEM-907, Omron, Osaka, Japan) [15].

A blood sample was collected from each participant by venepuncture after an overnight fast of at least 8 h. The SUA, creatinine, serum lipids and fasting plasma glucose were tested in the laboratory of the General Hospital of the Chinese People’s Liberation Army. The SUA and creatine were measured by oxidization with the specific enzyme uricase.

The eGFR was calculated according to the equation developed by the adaptation of the modification of diet in renal disease (MDRD) equation, based on data from Chinese CKD patients [16], which is written as:For male, eGFR = 175 × Scr^−1.234^ × age^–0.179^
For female, eGFR = 175 × Scr^−1.234^ × age^–0.179^ × 0.79

Above, Scr is the serum creatinine in mg/dl and age is in years.

### 2.3. Definitions

Hyperuricemia was defined as an SUA level higher than 360 μmol/L (~6 mg/dl) in women and 420 μmol/L (~7 mg/dl) in men [17]. There is a debate regarding the most appropriate cutoff for the eGFR in identifying reduced kidney function. The two cutoffs most often used to exclude individuals from reference interval studies are <60 mL/min/1.73 m^2^ and <90 mL/min/1.73 m^2^ [18]. Some studies supported the notion that the use of 90 mL/min/1.73 m^2^ can be more beneficial in regard to health outcomes [19]. As the early identification of kidney dysfunction using a higher cutoff value may have greater public health significance, in this study, we used 90 mL/min/1.73 m^2^ to define a reduced eGFR.

Hypertension was defined as a systolic BP of 140 mmHg or higher, a diastolic BP of 90 mmHg or higher, the use of any antihypertensive medication or any self-reported history of diagnosed hypertension [20]. Dyslipidemia was defined according to the Chinese guidelines for the management of dyslipidemia in adults [21]. Diabetes was defined as self-reported or ever-diagnosed disease or as a fasting plasma glucose (FPG) over 7.0 mmol/L. The definitions of cigarette smoking and alcohol drinking were consistent with our previous publications [3,15]. BMI was categorized into three groups: under/normal weight (BMI < 24 kg/m^2^), overweight (BMI ≥ 24 kg/m^2^ but <28 kg/m^2^) and obesity (BMI ≥ 28 kg/m^2^) [22]. FMR was calculated as the fat mass in kilograms divided by the muscle mass in kilograms.

### 2.4. Statistical Analyses

After excluding individuals with missing values for the FMR, SUA and eGFR, the final analytic sample included 31171 participants. Continuous data were presented as means with standard deviations (SDs), except for the eGFR, which was presented as the median (interquartile range, IQR) because of its high skew. Categorical variables were shown as frequencies and proportions. For descriptive purposes, the basic characteristics of the study population were presented according to presence or absence of HUA and reduced eGFR. The LMS (lambda, mu, sigma) method was used to construct the sex-specific growth curves of the FMR with advanced age [23].

The FMR was classified into four groups according to its centile distribution in each sex, i.e., the Q1 (less than P_25_), Q2 (P_25_–P_49_), Q3 (P_50_−P_74_) and Q4 (P_75_ and above). We analyzed the associations of the FMR with HUA and reduced eGFR using logistic regression models, in which sampling clusters and strata were considered using the Surveylogistic procedure in the statistical software SAS. Multivariable-adjusted odds ratios (ORs) were reported with their 95% confidence intervals (CIs). To avoid reverse causality resulting from the declined kidney function accompanied by significant muscle waste in senior participants, the association analyses of the FMR and reduced eGFR were restricted to people aged 20–59 years.

The covariates included in the multivariable logistic regression models varied based on different study aims. Several models were developed. Model 1 was adjusted for the demographic, health-related lifestyle and personal disease history information. Model 2 was additionally adjusted for BMI to explore the effect of the FMR independently from BMI. In the analyses of the association between the FMR and reduced eGFR, another model, Model 3, was designed to adjust for the HUA status in order to explore whether there was a modified effect of HUA on the association. We also performed age-stratified analyses to investigate whether the associations varied based on age.

Given the associations between the FMR, HUA and reduced eGFR, we performed a mediation analysis to further explore whether there was an indirect effect of HUA on the association between the FMR and reduced eGFR. The methodology and hypothesis were described in detail in our previous study [24]. In the mediation analysis, the FMR was categorized into two groups based on its sex-specific cutoff values by calculating the Youden Index (the maximum value of sensitivity + specificity − 1), yielded by the covariates’ adjusted receiver operator characteristic (ROC) curves. We further investigated the association between HUA and reduced eGFR using multivariable logistic regression models, and the results are presented in the Appendix A.

All *p* values were two-sided, and a *p* value of less than 0.05 was considered to be statistically significant. Analyses were performed with SAS (version 9.4, SAS Institute Inc., Cary, NC, USA). The GAMLSS package in R (version 4.0, R Core Team, Vienna, Austria) was used to perform the LMS method. As there were substantial sex-based differences in the FMR, HUA and eGFR, we performed all the analyses accounting for variations by sex.

## 3. Results

### 3.1. Basic Characteristics

The general characteristics of the study population are summarized in Table 1. Among the overall 31171 participants, 40.2% were male, with an average age of 48.71 ± 13.36 years. HUA cases were predominant among males (24.65% vs. 11.56% in women), people living in the urban areas (18.83% vs. 13.27% in rural areas), individuals with a higher educational level (18.59% in the college groups vs. 15.46% in the elementary/lower group), ever-smokers (23.81% vs. 13.93% in never-smokers), ever-drinkers (23.17% vs. 12.51% in never-drinkers), overweight or obese people (24.93% vs. 9.47% in the under/normal weight group) and participants in higher FMR categories (29.39% in the Q4 group vs. 6.48% in the Q1 group). Similar effects of the age trend and urban–rural and educational disparities on the prevalence of reduced eGFR were observed. Men, ever-smokers, alcohol drinkers, overweight/obese people and individuals with a higher FMR also had a higher prevalence of reduced eGFR. The age- and sex-specific prevalence of HUA and reduced eGFR are summarized in Appendix A.

### 3.2. Fat-to-Muscle Ratio and Its Association with Hyperuricemia

The sex-specific growth curves of the FMR are presented in Figure 1. The FMR slightly increased with advanced age in women, but in men, the FMR increased from early adulthood to middle age and was then preserved until senior age. The overall and age-stratified associations between the FMR and HUA are presented in Figure 2 and Appendix A. Model 2 shows that the effect of the increased centile groups of FMR on the risk of HUA was independent from the BMI. The forest plots revealed that people in higher FMR categories were more likely to have HUA. Compared with the Q1 of FMR, the ORs (95% CI) of the Q2, Q3 and Q4 groups in men were 1.60 (1.32–1.95), 2.31 (1.91–2.80) and 2.71 (2.15–3.43), respectively. In women, the magnitudes were larger, and the ORs of the Q2, Q3 and Q4 FMR groups were 1.91 (1.56–2.34), 2.67 (2.12–3.36) and 4.47 (3.40–5.89), respectively.

The age-stratified analyses revealed that, in men, the ORs of Q2–Q4 of FMR were similar between different age groups, but in women, the Q4 FMR group seemed to have a greater effect among younger females. Compared with the Q1 FMR group, Q4 group had an OR of 6.27 (3.35–11.73) among the group of participants aged <40, and we observed ORs of 4.29 (2.86–6.41) and 4.55 (2.62–7.89) in the groups aged 40–59 and 60 or above, respectively. In both the overall and the age-stratified analyses, the highest FMR group, Q4, had a greater effect among females than their male counterparts. Further details are available in Figure 2 and Appendix A.

### 3.3. The Association between FMR and Reduced eGFR

The associations between the FMR and reduced eGFR are presented in Figure 3 and Appendix A. In men, the magnitude of the ORs became smaller when HUA was adjusted in the regression models. The ORs (95% CI) of Q2, Q3 and Q4 of FMR were 1.26 (1.03–1.54), 1.19 (0.93–1.52) and 1.22 (0.92–1.63), respectively. However, in females, the FMR was not found to be associated with reduced eGFR in any FMR category. The positive associations in Model 1 and Model 2 disappeared when we adjusted for BMI and HUA in Model 3.

The age-stratified analyses revealed a possible modification effect of age on the associations in men. The FMR was positively associated with reduced eGFR in the older men (aged 40–59 years) but not in men younger than 40. Notably, in women, after adjusting for BMI and HUA, there was no positive association between the FMR and reduced eGFR in different age groups (Figure 3 and Appendix A).

### 3.4. The Mediation Analysis of the Association between FMR and Reduced eGFR

HUA had strong effect on the reduced eGFR, and the associations between HUA and reduced eGFR are shown in Appendix A. As no positive association between the FMR and reduced eGFR was observed in females, we only applied the mediation analysis to male participants. The result showed that HUA mediated the association between the FMR and reduced eGFR remarkably, and the indirect effect accounted for 35.11% of the total effect. The OR (95% CI) of the indirect effect was 1.08 (1.05–1.10) (Table 2).

## 4. Discussion

In this representative sample of Chinese adults, the FMR was found to be independently associated with hyperuricemia and reduced eGFR, and the associations varied based on both sex and age. Furthermore, the effect of the FMR on reduced eGFR was significantly mediated by hyperuricemia in men.

Some studies have documented the FMR as an indicator for identifying members of populations at a high risk of developing cardiovascular diseases and its risk factors [11,12,13,25,26]. Nevertheless, to the best of our knowledge, this is the first study that explored the associations of the FMR with HUA and reduced eGFR in Chinese adults. The primary findings of this study suggest that the FMR can be used as an indicator of HUA independently from the traditional excess adiposity indexes, such as BMI, and a higher FMR is also associated with a reduced eGFR. In addition, by deconstructing the total effect of the FMR on reduced eGFR, our study revealed indirect associations between the FMR, HUA and reduced eGFR.

In recent years, the function of the skeletal muscle and the balance of MM and FM have raised great concern. Termed as “sarcopenia” and “sarcopenic obesity”, the loss of muscle mass and severe imbalance between the distribution of muscle and fat tissue with ageing are believed to be associated with various health hazards [27]. The associations of the FMR with HUA and reduced eGFR highlight the importance of exploring alternative indicators to BMI and focusing our attention on the balance of the skeletal muscle and fat mass in the early identification of the individuals at a high-risk of HUA and reduced eGFR in community-dwelling populations.

In contrast with traditional adiposity indicators, such as the BMI, body fat percentage or waist circumference, the FMR can reflect the relative relationship and imbalance between the skeletal muscle and fat mass. The decline in skeletal MM accompanied by an increase in FM may lead to a dual metabolic burden, which leads to a high risk of developing cardiometabolic disorders [25]. Although the precise mechanism of the relationship between the FMR and HUA is not clear, previous studies have shown that an increased FMR is closely related to degradation of insulin and chronic inflammation and can result in impaired cardiometabolic functions [28,29]. These shared factors may contribute to research demonstrating the relationship between FMR and HUA.

Sex differences in the trajectory of the FMR with ageing were observed. The decrease in MM due to enhanced catabolism has been suggested to be combined with obesity, thereby resulting in changes in the FMR [30]. Compared with men, women had an increased FMR with ageing, and this may partially explain the stronger effect of the FMR on HUA in females in the current study. Another possible reason is that there is a sex-based disparity in the risk of the onset of HUA during ageing, with the risk declined in men but increased in women [3,9].

Overweight and obesity have been suggested to increase the risk of kidney dysfunction in previous studies. Madero et al. reported that adiposity indicators, such as the BMI and waist circumference, were associated with kidney function decline [28]. Wang et al.’s study supported the notion that the combined effects of dyslipidemia and high adiposity were significantly associated with the decline in the eGFR in a middle-aged Chinese population, especially in men [31]. Impaired insulin resistance, chronic inflammatory reactions, lipid toxicity and the chronic and excessive activation of the sympathetic nervous system may contribute to the development of kidney dysfunction and, eventually, CKD [32]. The analyses of body composition indicators other than those that only represent the fat mass have brought new insights on our understanding of the prevention of kidney dysfunction. Kim et al. investigated the effects of the fat mass and muscle mass index (MMI, muscle divided by height) on CKD and revealed that a higher FM and lower MMI, but not BMI, were associated with a higher risk of CKD among Korean older adults [33]. The findings of our study also support the notion that men with a higher FMR are more likely to have a reduced eGFR, even after the adjustment for BMI. However, this association was not observed in females. The effects of sex-based differences and disparities on the prevalence and progression of kidney dysfunction may be related to the direct effects of sex steroids on the kidneys, sex differences in NO metabolism and oxidative stress and the gender-differential impacts of comorbidities and lifestyle risk factors [34]. The reason underlying the sex disparity observed in this study is still not clear but may be related to the competing risk of more predominant effects caused by increased fat mass over the course of aging in women. The age-variable associations between the FMR and reduced eGFR in men highlight the importance of focusing our attention on the senior members of society, among whom sarcopenic obesity or sarcopenia occurs more frequently.

Although hyperuricemia is believed to be a major contributor to the progression of kidney dysfunction, the role of HUA in the development of CKD was firstly viewed as one that can be solely attributed to the retention of SUA caused by the GFR fall [35]. Although some Mendelian randomization studies have found no evidence of the casual role of serum urate in CKD [36], a great number of epidemiological, experimental and clinical studies have suggested that uric acid has an important pathogenic role in kidney disease [4]. The potential mechanism underlying this procedure may be related to inflammation, endothelial dysfunction and the activation of the renin-angiotensin system [1]. Clinical practice has also suggested that uric-acid-lowering therapy can benefit people by slowing the progression of CKD [37]. In addition to the strong associations between HUA and kidney function, our findings also revealed that HUA may play a mediating role in the effect of the FMR on reduced eGFR. This finding suggests that the management of HUA may bring additional health benefits aiding in the prevention of reduced eGFR. Although the cross-sectional design of our study cannot provide information on the time sequence of HUA and reduced eGFR in the study population, the analysis still has value for our understanding of the effects of muscle health and body composition imbalance on reduced kidney function, as well as its prevention and management.

A major strength of our study is that the data were derived from a representative national survey, with a high degree of heterogeneity. Hence, it provides a unique dataset that enabled us to explore the effect of the fat-to-muscle ratio, which is a novel indicator that represents skeletal muscle health and the balance between the FM and MM, as well as their effects on various health outcomes. In addition, the training programs and the on-site quality control strategies used here ensured the credibility of the results. The limitations of our study should also be acknowledged. First, it should be recognized that the use of eGFR as an estimation of kidney function should be approached with caution. As previous studies have demonstrated, dietary factors, such as meat consumption, can influence the creatine levels [14]. Moreover, as blood creatine is predominantly derived from muscle, the estimation of the GFR based on creatine may lead to underestimated kidney function measurements among senior people and, in contrast, overestimation among younger people, especially in the case of muscular young men [14]. Nevertheless, as we limited the analysis of the eGFR to people aged 20–59, the direction of the positive association between the FMR and eGFR in the senior group tends toward the null hypothesis. Thus, it does not lead to a reversed conclusion. Secondly, it is true that the varied estimation formulas used to calculate the GFR, and the “one-off” testing of the serum creatine level, with no 3-month observation, which was used to calculate the eGFR, may have caused a misclassification in this study. Additionally, diet and other environmental factors may have affected the measurements of the level of serum creatine. Nevertheless, as the purpose of this study was the association exploration, rather than the prevalence or the disease burden estimation, these limitations are acceptable and would not reverse the study conclusions. Thirdly, as mentioned above, the nature of the study’s cross-sectional design limited the causal inference of the exposure and health outcomes, and future longitudinal data are still needed in order to perform a validation analysis.

## 5. Conclusions

In summary, our study, for the first time, investigated the possible role of the fat-to-muscle ratio as a novel indicator for identifying individuals at a high risk of HUA and reduced eGFR in a representative Chinese adult population. The FMR was found to be positively associated with HUA and reduced eGFR, with the additional effects of sex and age differences. People with a higher FMR were more likely to have HUA in both sexes and were more likely to have a reduced eGFR in the case of middle-aged men. HUA may act as a mediator in the association between the FMR and reduced eGFR in men. The FMR should be used in community health screening for, and the co-management of, elevated serum uric acid, which should be considered in kidney dysfunction prevention strategies.

## Figures and Tables

**Figure 1 nutrients-14-04193-f001:**
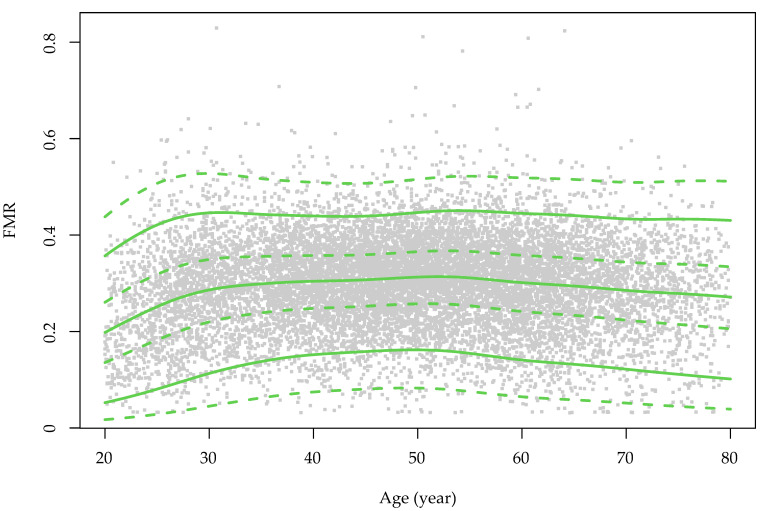
The sex-specific growth curve of the fat-to-muscle ratio over the course of ageing in the study population. FMR: fat-to-muscle ratio. The green color is for males and red is for females. The grey dots represent the distribution of the FMR values for each sex.

**Figure 2 nutrients-14-04193-f002:**
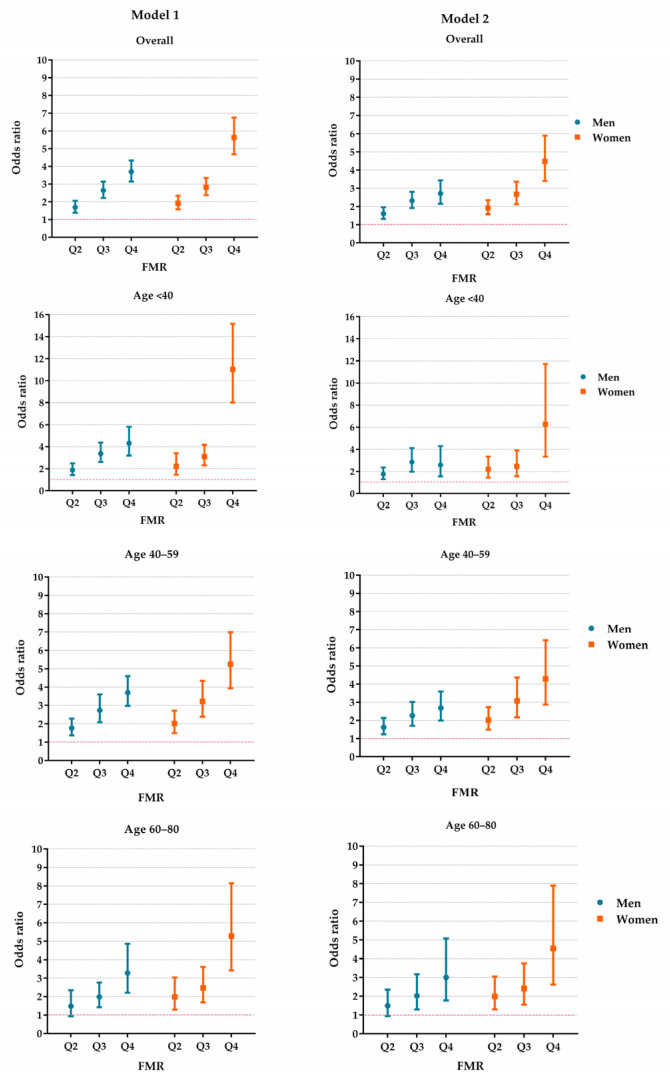
The effect of FMR on HUA in the study population. Model 1 was adjusted for age, rural/urban location, educational level, study sites, alcohol drinking status (male only), smoking status (male only), hypertension, diabetes, dyslipidemia and serum creatine. Model 2 was additionally adjusted for body mass index based on Model 1. FMR: fat-to-muscle ratio. HUA: hyperuricemia. Q2–Q4: the values fall in the 25–49th, 50–74th and 75–100th centiles of FMR.

**Figure 3 nutrients-14-04193-f003:**
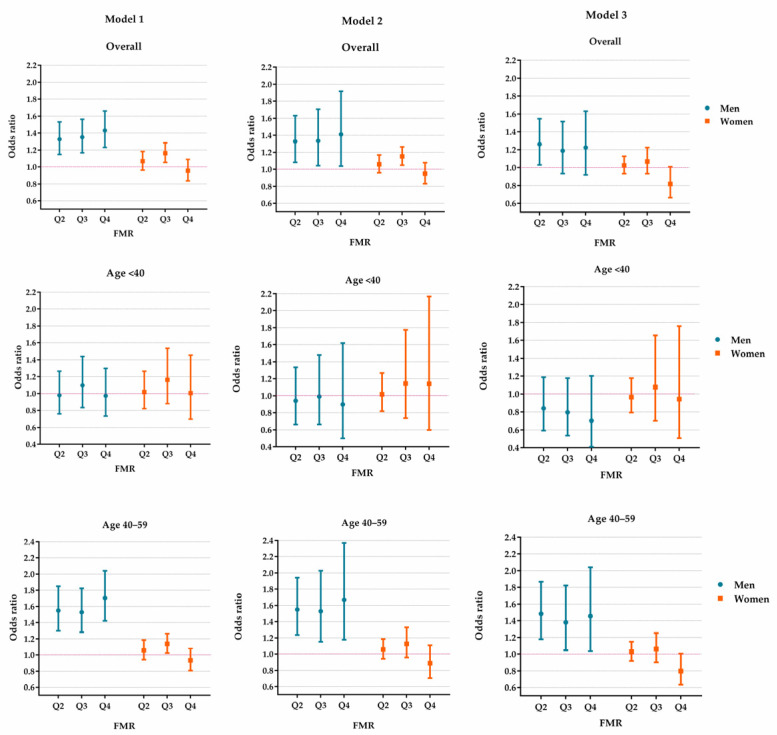
The effect of FMR on reduced eGFR in the study population aged 20–59. Model 1 was adjusted for age, rural/urban location, educational level, alcohol drinking status, smoking status, hypertension, diabetes and dyslipidemia. Model 2 was additionally adjusted for body mass index based on Model 1. Model 3 was additionally adjusted for hyperuricemia based on Model 2. FMR: fat-to-muscle ratio. Q2–Q4: the values fall in the 25–49th, 50–74th and 75–100th centiles of FMR.

**Table 1 nutrients-14-04193-t001:** Basic characteristics of the study population (*n* = 31171).

	Men (*n* = 12523)								Women (*n* = 18648)							
	HUA		Non-HUA		Reduced eGFR		Non-Reduced eGFR		HUA		Non-HUA		Reduced eGFR		Non-Reduced eGFR	
**Age (*n*, %)**																
20−	371	12.02	913	9.66	164	3.80	1120	13.66	156	7.24	1846	11.19	132	2.29	1870	14.52
30−	596	19.30	1402	14.86	438	10.13	1560	19.02	197	9.14	2852	17.29	418	7.24	2631	20.43
40−	798	25.85	2368	25.10	1001	23.16	2165	26.40	375	17.39	4612	27.97	1300	22.53	3687	28.63
50−	706	22.87	2419	25.64	1161	26.86	1964	23.95	646	29.96	3987	24.18	1727	29.92	2906	22.57
60−	446	14.45	1705	18.07	1050	24.29	1101	13.42	553	25.65	2426	14.71	1524	26.41	1455	11.30
70–80	170	5.51	629	6.67	508	11.76	291	3.55	229	10.62	769	4.66	670	11.61	328	2.55
**Urban (*n*, %)**	2208	71.53	5637	59.74	3144	72.74	4701	57.32	1542	71.52	10525	63.82	4164	72.15	7903	61.37
**Rural (*n*, %)**	879	28.47	3795	40.22	1176	27.21	3498	42.65	613	28.43	5959	36.13	1602	27.76	4970	38.60
**Education (*n*, %)**																
Illiterate/elementary school	437	14.16	1821	19.30	840	19.44	1418	17.29	783	36.32	4851	29.41	2173	37.65	3461	26.88
High school	1519	49.21	4909	52.02	2101	48.61	4327	52.76	962	44.62	7611	46.15	2538	43.98	6035	46.87
College or higher	1123	36.38	2696	28.57	1373	31.77	2446	29.83	407	18.88	4005	24.28	1053	18.25	3359	26.09
**Smoking (*n*, %)**																
Never	1005	32.56	2885	30.57	1370	31.70	2520	30.73	2070	96.01	16112	97.70	5597	96.98	12585	97.73
Quit	541	17.53	1493	15.82	912	21.10	1122	13.68	28	1.30	73	0.44	40	0.69	61	0.47
Current	1538	49.82	5057	53.59	2038	47.15	4557	55.57	58	2.69	304	1.84	134	2.32	228	1.77
**Alcohol drink (*n*, %)**																
Never	547	17.72	2583	27.37	1086	25.13	2044	24.92	1767	81.96	13602	82.48	4733	82.01	10636	82.60
Quit	319	10.33	955	10.12	596	13.79	678	8.27	55	2.55	317	1.92	168	2.91	204	1.58
Current	2220	71.91	5878	62.29	2636	60.99	5462	66.60	334	15.49	2561	15.53	867	15.02	2028	15.75
**BMI (mean, SD)**	25.89	3.50	23.92	3.37	24.71	3.27	24.25	3.61	25.81	3.76	23.38	3.44	24.12	3.38	23.45	3.62
<24 **(*n*, %)**	863	27.95	4882	51.74	1793	41.49	3952	48.19	685	31.77	9921	60.15	2950	51.12	7656	59.45
24− **(*n*, %)**	1457	47.20	3516	37.26	1866	43.17	3107	37.89	921	42.72	4962	30.09	2073	35.92	3810	29.59
28− **(*n*, %)**	767	24.85	1038	11.00	663	15.34	1142	13.92	550	25.51	1609	9.76	748	12.96	1411	10.96
FM (Mean, SD)	18.17	6.06	14.55	5.74	16.08	5.61	15.11	6.21	23.32	7.21	18.77	6.27	20.14	6.25	18.92	6.64
MM (Mean, SD)	53.20	6.36	50.52	5.87	51.18	5.97	51.19	6.18	37.35	3.55	36.43	3.42	36.54	3.44	36.54	3.45
**FMR (*n*, %)**																
Q1	337	10.92	2780	29.46	804	18.60	2313	28.20	167	7.75	4491	27.23	1057	18.32	3601	27.96
Q2	617	19.99	2529	26.80	1097	25.38	2049	24.99	348	16.14	4308	26.12	1362	23.60	3294	25.58
Q3	917	29.70	2215	23.48	1176	27.21	1956	23.85	568	26.34	4105	24.89	1642	28.45	3031	23.54
Q4	1216	39.39	1912	20.26	1245	28.81	1883	22.96	1073	49.77	3588	21.76	1710	29.63	2951	22.92
SUA (mean, SD)	484.03	58.54	330.69	55.40	397.94	88.47	353.00	81.65	410.30	50.72	262.30	49.86	311.20	72.92	265.19	61.81
eGFR (M, IQR)	89.57	24.28	99.40	24.69	80.64	11.90	105.78	19.57	85.92	26.80	101.49	27.50	80.63	12.20	108.49	22.63

HUA: hyperuricemia. eGFR: estimated glomerular filtration rate, mL/min per 1.73 m^2^. BMI: body mass index, kg/m^2^. FM: fat mass, kg. MM: muscle mass, kg. FMR: fat-to-muscle ratio. Q1–Q4: the numbers fall in the 0–24th, 25–49th, 50–74th and 75–100th ranges of FMR. SUA: serum uric acid, μmoL/L. SD: standard deviation. M: median. IQR: interquartile range. *n* represents the numbers in certain category and % refers to the column proportions.

**Table 2 nutrients-14-04193-t002:** The mediation analysis of hyperuricemia’s effect on the association between the FMR and reduced eGFR among men aged 20–59.

	OR	95% CI	
Total effect	1.255	1.084	1.426
Natural direct effect (NDE)	1.166	1.008	1.323
Natural indirect effect (NIE)	1.077	1.052	1.102
% Mediated	35.11	14.98	55.24

FMR: fat-to-muscle ratio; eGFR: estimated glomerular filtration rate; OR: odds ratio; CI: confidence interval.

## Data Availability

The datasets generated or analyzed in this study are available from the corresponding author upon reasonable request.

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
