# Peer review of "Fat-to-Muscle Ratio Is Independently Associated with Hyperuricemia and a Reduced Estimated Glomerular Filtration Rate in Chinese Adults: The China National Health Survey"

_nutrients, 2022, doi:10.3390/nu14194193_

Round 1
Reviewer 1 Report
The major methodological flaw results from the inappropriate use of definitions.
The definition of CKD is precise and says about eGFR below 60ml/min/1.73m2 for at least 3 months or normal/increased eGFR value in the presence of other signs of kidney damage (e.g. proteinuria), present for at least 3 months.
There is no data on 3-month observation or any parameters confirming any kind of kidney damage. Moreover, eGFR values decrease with age. Therefore, most of the people over 60 have their eGFR below 90, but it doesn’t mean they have CKD.
Taking into account these physiological conditions, one should not create a cohort out of the patients with such a range of age – from 20 to 80 years.
The connections between hyperuricemia and obesity, hyperuricemia and CKD, as well as between obesity and CKD, are indisputable. However, the way the results are presented suggests also that FMR has an effect on CKD. It is, in fact, the opposite, because patients with CKD suffer from decreased muscle mass, not vice versa. In fact, the lower the muscle mass, the lower serum creatinine concentration. So the statement about “ the effect of FMR on CKD” has no rationale, “the effect of CKD on FMR” should rather be analyzed.
Author Response
Dear professor,
We appreciate your valuable comments very much, which provide honest and great information for us to improve the manuscript. We are honored to have this opportunity to make a response here and have revised the paper substantially based on your suggestions. Your concern is quite right that using a different definition of CKD in the current manuscript may result in confusion and misleading to the potential readers. To avoid this confusing understanding, we replaced “CKD” with “reduced renal function” to make it clearer and more appropriate. We hope to focus on an earlier progression stage of CKD, which is the “reduced renal function”, to achieve early identification of high-risk population and risk factors that may contribute to the prevention of CKD or more severe kidney diseases.
As mentioned in the method section, CNHS is a community-based health survey, so people recruited in this study mainly came from natural living communities but not hospitals. By cross-sectional design, data were collected based on one time field survey, so the 3-month observation or kidney damage information was not available. I realize that this is one of the most common limitations for epidemiological studies, not only for CKD, but also other diseases, such as the diagnosis of hypertension, diabetes, etc. We have recognized this limitation and have discussed this issue in the last part of the paper as” It is true that varied estimation formula used to calculated GFR and the “one-off” testing of serum creatine level, which was used to calculate eGFR, may cause misclassification in this study. And there could be diet and other environmental factors that are not related to kidney disease have effect on the level of serum creatine level. Nevertheless, as the purpose of this study focuses on the association exploration, rather than the prevalence or the disease burden estimation, the mentioned limitations could be acceptable and would not reverse the study conclusions.” We sincerely appreciate your understanding and will use the word “CKD” more carefully in the future.
To avoid the possible inverse causality, we have restricted the analytic sample in people aged 20-59, when estimating the association between FMR and reduced renal function. The relevant tables and figures were revised, and the discussion was also revised based on the updated results.
We appreciate your valuable comments which help us to improve our understanding of the current study topic and also provide us great enlightens for our future work. We sincerely ask for your favor to help us improve the quality of the revised paper and looking forward to your suggestions.
Reviewer 2 Report
This interesting study shows the association between the biopohysical parameter FMR and kidney function. I have the following remarks:
- Had it been documented what the causes were for CKD in each patient? This could partly explain the differences seen in men and women.
- Is there anything known concerning different grades of CKD (including end-stage renal disease)?
Author Response
Dear professor,
Many thanks for your comments and we are honored to have this opportunity to make a response to your concern.
As in the response to another reviewer, CNHS is a community-based health survey, so people recruited in this study mainly came from natural living communities. As suggested by the first professional reviewer, we have replaced “CKD” with “reduced renal function” to make clearer understanding. Unfortunately, we did not collect specific causes of CKD as most of the participants were asymptomatic, thus the number of end-stage renal disease is even rare. Nevertheless, by literature review, we have added relevant discussion on the possible mechanism for the sex disparities on reduced renal function. Please see in lines 322-327. Perhaps in the future we could conduct hospital-based survey focusing on CKD patients, so that the associated factors with different grades of CKD can be estimated.
We appreciate your effort and help very much.
Reviewer 3 Report
This paper is written in a clear style and data analysis is sound.
Minor comments:
1. The formula in line 120 is not very clear. Please provide separate formulas for men and women. Letter type is too small.
2. Can the authors define the ethnicity of the study subjects? There are different Chinese ethnic groups.
3. Please provide more details on the quantification of the fat-to-muscle ratio and how the technique was validated (in the literature) by independent approaches.
4. An independent effect implies statistical significance in a multivariable model in which adjustments were made for a series of covariables that may be confounding factors. However, there may be hidden confounding factors. All models are wrong but some are useful.
Author Response
Dear professor,
Thank you very much for your valuable comments and we are honored to have this opportunity to have a response here.
- We have provided separate formulas by sex and also revised the presentation of them. Please see in the method section, lines 124-126.
- Thank you. In this study, only Han Chinese were included, so there is only one ethnicity in the study population. We have described the selection of population for the analytic sample in the method section, please see in lines 99-100. Thanks for this concern.
- Thank you. We have added the description of the calculation of FMR in the method section, please see in lines 146-147. There are previously studies used FMR as an alternative indicator of excess adiposity and the imbalance of fat mass and skeletal muscle in diverse populations. For instants, Ramírez-Vélez et al’s study which published in Nutrients 2018, used bioelectric impedance assessment (BIA), the same measurement with our study, to collect body composition information; In Eun et al’s study (Eun et al. Int J. Med. Sci. 2021), BIA method was also used to measure fat mass and muscle mass in adults. In fact, BIA has been proved to be a simple, cost-effective, and non-invasive tool for body composition measurement and can be applied across a wide range of populations.
- Many thanks for your suggestions on this topic, we totally agree with you that it is impossible to include all the confounding factors in the regression model. Besides those we still don’t know, even the known risk factors are hardly to be considered comprehensively because of the limitation in resource and funding. We should use the word “independent” very carefully, and have reduced the use of “independent” in the revised paper, e.g., in the figure legends. Nevertheless, in this manuscript, we only would like to demonstrate that FMR is associated with HUA and renal function independently from BMI, but not all the other factors. Therefore, in this scenario, we hope that the wording of “independently” is acceptable and applicable. Thank you very much for your understanding and help.
Reviewer 4 Report
There are many studies focused on the link between hyperuricemia and CKD, but there are many gaps to be filled in order to establish this association, therefore your study represents an important step forward to understand the possible incriminated pathophysiological mechanisms. In my opinion, your article is suitable for publication.
Author Response
Dear professor,
We gratefully appreciate your comments and are encouraged to carry out more related scientific research in the future. Many thanks for your time and efforts.
Round 2
Reviewer 1 Report
“Renal function” is a general term which includes glomerular, tubular, endocrine and metabolic activities of this organ.
The value of glomerular filtration rate is the only aspect of kidney function the Authors test, so it would be appropriate to use the term “reduced eGFR” instead of “reduced RF”.
Once again, there is no rationale for diagnosis of CKD in the analyzed population, so some restraint in the Introduction and Discussion, focusing on CKD, should be applied. Instead, cautious assumption of CKD as a probable reason for eGFR decline, may be put into the text. The fact that no 3-month observation is available, should be commented on and included into study limitations.
Presentation of the results requires correction.
Table 1 – there is no description of presented values, e.g. two columns for HUA – what do they represent – number of patients? Percentage? Mean values? SD?
If IQR is given, there should be a range = two values (…-…) instead of one.
Figure 1. Sex-specific changes are shown on two separate graphs, but there is no description which colour (green or red ) stands for which sex.
Extensive English editing is required.
Phrases in lines 41-42, 55-57, 71, 83, 157-159, 281, 292-293, 299, 301, 320-322, etc. should be rewritten.
Terms like “causality caused”, “decomposition of body composition”, “the associations varies on sex”, should be rephrased.
Author Response
Dear professor,
Thank you very much for your valuable comments. We appreciate your considerable time and effort in helping us improve the paper and are feel lucky to have this opportunity learn from your expertise. We have revised the manuscript substantially according to your suggestions, and are honored to have this chance to make a response here.
Thank you for your suggestion on the appropriate representation of our study. We have removed “reduced renal function” and use “reduced eGFR” instead throughout the manuscript. Many thanks for your help, that avoid further misleading to the potential readers of this paper. We have learned and will choose the terminology more carefully in our future research.
As you suggested, it is not appropriate to use CKD so much in the introduction or in the discussion part, so we revised these two parts, updated some references and remove some irrelevant references. Thanks to your suggestion, we looked deeper into the limitation of eGFR on the estimate of GFR, and have added description in the manuscript both in the introduction and the discussion sections. Please allow us to quote here for your convenience:
In the introduction, we described why eGFR was used to estimate GFR in this epidemiological study: “Glomerular filtration rate (GFR) is the ideal overall index of kidney function, but the measurement is cumbersome and impractical for general detection and management of kidney disease (O'Riordan, P et al. BMJ 2014). In this case, estimated GFR (eGFR) by equations is widely used as a surrogate of GFR, especially in large-scale epidemiological studies.”
In the discussion part, we discussed the limitations of using eGFR and the unavailable data on the 3-month observation: “The limitations of our study should also be acknowledged. First, it should be recognized that the use of eGFR as an estimation of kidney function should be with caution. As previous studies have demonstrated that dietary factors, such as meat consumption, will influence the creatine level (O'Riordan, P et al. BMJ 2014). Moreover, as blood creatine is predominantly derived from muscle, the estimation of GFR based on creatine may lead to underestimated kidney function among senior people but overestimation among younger people, especially those muscular young men (O'Riordan, P et al. BMJ 2014). Nevertheless, as we limited the analysis related to eGFR in people aged 20-59, the direction of the positive association between FMR and eGFR in the senior group tends to toward the null hypothesis, thus will not lead to a reversed conclusion. Second, it is true that varied estimation formula used to calculated GFR and the “one-off” testing result of serum creatine level with no 3-month observation, which was used to calculate eGFR, may cause misclassification in this study.”
We are sorry for the unclear presentation in the results section, and have revised the relevant part based on your concerns. In table 1, we have added footnotes explaining the meaning of n and % in the table. For the use of IQR, we checked the definition again, which is as follows:
The interquartile range defines the difference between the third and the first quartile. Quartiles are the partitioned values that divide the whole series into 4 equal parts. So, there are 3 quartiles. First Quartile is denoted by Q1 known as the lower quartile, the second Quartile is denoted by Q2 and the third Quartile is denoted by Q3 known as the upper quartile. Therefore, the interquartile range is equal to the upper quartile minus lower quartile.
Interquartile range = Upper Quartile – Lower Quartile = Q3 – Q1
According to the definition of IQR, the values we presented is appropriate. Thank you for your understanding.
We are sorry for the unclear description of figure 1, and have added relevant demonstration on colors and dots, i.e., green is for men and red is for women, the grey dots represent the values of FMR in each sex.
We are very sorry for the grammar errors and language problems in this manuscript. We have tried our best to ask an export to help us to edit the paper thoroughly. We hope that this version is acceptable.
We would like to gratefully express our appreciation again for your help and sincerely looking forward to your help and endorsement.